# Molecular Mechanism of Pancreatic β-Cell Failure in Type 2 Diabetes Mellitus

**DOI:** 10.3390/biomedicines10040818

**Published:** 2022-03-31

**Authors:** Hideaki Kaneto, Tomohiko Kimura, Masashi Shimoda, Atsushi Obata, Junpei Sanada, Yoshiro Fushimi, Taka-aki Matsuoka, Kohei Kaku

**Affiliations:** 1Department of Diabetes, Endocrinology and Metabolism, Kawasaki Medical School, 577 Matsushima, Kurashiki 701-0192, Japan; tomohiko@med.kawasaki-m.ac.jp (T.K.); masashi-s@med.kawasaki-m.ac.jp (M.S.); obata-tky@med.kawasaki-m.ac.jp (A.O.); gengorou@med.kawasaki-m.ac.jp (J.S.); fussy.k0113@med.kawasaki-m.ac.jp (Y.F.); kka@med.kawasaki-m.ac.jp (K.K.); 2The First Department of Internal Medicine, Wakayama Medical University, 811-1 Kiimidera, Wakayama 641-8509, Japan; matsuoka@wakayama-med.ac.jp

**Keywords:** PDX-1, MafA, incretin reeceptor, GLP-1 receptor activator, coronavirus infection

## Abstract

Various important transcription factors in the pancreas are involved in the process of pancreas development, the differentiation of endocrine progenitor cells into mature insulin-producing pancreatic β-cells and the preservation of mature β-cell function. However, when β-cells are continuously exposed to a high glucose concentration for a long period of time, the expression levels of several insulin gene transcription factors are substantially suppressed, which finally leads to pancreatic β-cell failure found in type 2 diabetes mellitus. Here we show the possible underlying pathway for β-cell failure. It is likely that reduced expression levels of MafA and PDX-1 and/or incretin receptor in β-cells are closely associated with β-cell failure in type 2 diabetes mellitus. Additionally, since incretin receptor expression is reduced in the advanced stage of diabetes mellitus, incretin-based medicines show more favorable effects against β-cell failure, especially in the early stage of diabetes mellitus compared to the advanced stage. On the other hand, many subjects have recently suffered from life-threatening coronavirus infection, and coronavirus infection has brought about a new and persistent pandemic. Additionally, the spread of coronavirus infection has led to various limitations on the activities of daily life and has restricted economic development worldwide. It has been reported recently that SARS-CoV-2 directly infects β-cells through neuropilin-1, leading to apoptotic β-cell death and a reduction in insulin secretion. In this review article, we feature a possible molecular mechanism for pancreatic β-cell failure, which is often observed in type 2 diabetes mellitus. Finally, we are hopeful that coronavirus infection will decline and normal daily life will soon resume all over the world.

## 1. A Variety of Pancreatic Transcription Factors Are Involved in the Development of the Pancreas and Differentiation of Endocrine Progenitor Cells into Mature Pancreatic β-Cells: Pancreas-Related Phenotype in Knockout Mice of Each Transcription Factor

Pancreatic islets are composed of α-, β-, δ-, ε-, and PP-cells, which secrete glucagon, insulin, somatostatin, ghrelin, and pancreatic polypeptide, respectively. A variety of pancreatic transcription factors are involved in the development of the pancreas and differentiation of endocrine progenitor cells into mature β-cells. Pancreatic and duodenal homeobox factor-1 (PDX-1) was identified by several independent research groups at around the same time. It is well known that PDX-1 plays a crucial role in the early stage of the development of the entire pancreas [1,2,3,4,5,6,7,8,9,10,11,12]. Hb9 plays an important role in the development of the dorsal pancreas [13,14] (Table 1). Arx, Isl-1, Pax4, Pax6, Nkx6.1 and Nkx2.2 are also involved in the development of the pancreas [15,16,17,18,19,20,21,22,23,24,25,26]. The phenotype in the pancreas in knockout mice of each transcription factor is as follows: Arx knockout mice, absence of α-cells and increase in β- and δ-cells [26]; Isl-1 knockout mice, absence of islet cells [15]; Pax4 knockout mice, absence of β-cells, decrease in δ-cells, and increase in α- and ε-cells [16,23]; Pax6 knockout mice, absence of α-cells, decrease in β-, δ- and PP-cells, increase in ε-cells [17,18,24]; Nkx6.1 knockout mice, decrease in β-cells; Nkx2.2 knockout mice, absence of β-cells, decrease in α- and PP-cells, and increase in ε-cells [19,20,23] (Table 1).

It is well known that PDX-1 plays a crucial role in the development of the whole pancreas [1,2,3,4,5,6,7,8,9,10,11,12], the differentiation of endocrine progenitor cells into mature β-cells [27,28,29,30,31,32,33,34,35,36,37], and maintenance of mature β-cell function [38,39,40,41,42,43,44,45]. PDX-1 is initially expressed in the gut region in the early stages of embryonic development. PDX-1 expression is preserved in endocrine progenitor cells during the development of the pancreas, but its expression is restricted to insulin-producing β-cells in the mature pancreas. In PDX-1 knockout mice, there was no pancreas [1], which clearly shows that PDX-1 plays a very important role during the process of pancreas formation (Table 1). Moreover, pancreatic agenesis is observed in subjects with loss of PDX-1 function [9]. In mature β-cells, PDX-1 transactivates several β-cell-related genes including insulin, GLUT2 and glucokinase [41,42]. Abnormal glucose metabolism, an increase in β-cell apoptosis and a decrease in islet mass were also observed in PDX-1 hetero-deficient mice [13,42].

NeuroD and neurogenin3 (Ngn3) function as transcription factors in the pancreas. NeuroD plays an important role in the development of the pancreas and in regulation as insulin gene transcription in mature β-cells [46,47,48,49,50,51,52]. It was reported that in NeuroD knockout mice, the β-cell number was markedly reduced, leading to severe diabetes mellitus and perinatal death [47] (Table 1). Neurogenin3 (Ngn3) is also involved in the differentiation of endocrine progenitor cells [51,52,53,54,55,56,57,58,59,60]. After bud formation, Ngn3 is transiently expressed in endocrine progenitor cells, and functions as a potential initiator of endocrine differentiation. It was reported that in transgenic mice overexpressing Ngn3, endocrine cell formation was markedly increased [52]. In contrast, it was reported that in Ngn3 knockout mice there were no endocrine cells. These findings clearly show that Ngn3 plays a crucial role in endocrine differentiation [53] (Table 1).

MafA was identified by several independent research groups around the same time. MafA transactivates insulin gene by binding the RIPE3b1 element, and its expression is observed only in β-cells [61,62,63,64,65,66,67,68,69,70,71,72]. Furthermore, abnormality of glucose metabolism was induced by MafA knockout [61]. In MafA knockout mice, insulin biosynthesis and glucose-stimulated insulin secretion were reduced (Table 1). These findings clearly show that MafA plays a crucial role in the maintenance of mature pancreatic β-cell function.

## 2. Reduced Expression Levels of Insulin Gene Transcription Factors Such as PDX-1 and MafA Are Involved in Pancreatic β-Cell Failure Found in Type 2 Diabetes Mellitus

Recently, obesity has markedly increased all over the world. Previously, it was thought that obesity was simple accumulation of fat tissues. However, it is now well known that obesity exerts different effects on our body, depending on the site of fat deposition. Obesity is the starting point of most metabolic diseases such as metabolic syndrome and type 2 diabetes mellitus. In subjects with obesity and/or metabolic syndrome, insulin resistance develops mainly due to overeating and/or lack of exercise, but sufficient insulin is secreted from intact β-cells to compensate for the insulin resistance. However, in subjects with insulin resistance, β-cells have no choice but to produce and secrete larger amounts of insulin, which finally leads to β-cell overwork. Additionally, β-cell function gradually deteriorates due to a large amount of free fatty acids and/or various inflammatory cytokines that are secreted from visceral fat tissues. This process is known as β-cell lipotoxicity. Such β-cell overwork and lipotoxicity finally lead to the development of type 2 diabetes mellitus in subjects with obesity and/or metabolic syndrome.

The major function of pancreatic β-cells is to secrete insulin when blood glucose levels are increased. However, when β-cells are exposed to chronic hyperglycemia after the onset of type 2 diabetes mellitus, β-cell function gradually deteriorates due to overwork for insulin biosynthesis and secretion. Once hyperglycemia becomes overt, β-cell function progressively deteriorates. Such β-cell failure is often observed in subjects with type 2 diabetes mellitus and is known as pancreatic β-cell glucose toxicity in clinical practice, as well as in the islet biology research area. In the diabetic state, hyperglycemia and the subsequently provoked oxidative stress suppress insulin biosynthesis and secretion and finally lead to apoptotic β-cell death [73,74,75,76,77,78,79,80,81,82,83,84,85]. This reduction in insulin biosynthesis and secretion is preserved by mitigating pancreatic β-cell failure with insulin preparation or SGLT2 inhibitors [86,87,88,89,90]. Additionally, an important concept regarding β-cell failure was recently proposed. It was shown that the reduction in β-cell mass was not only due to apoptotic β-cell death but also due to differentiation of insulin-producing mature β-cells into Ngn3-expressing endocrine progenitor cells [54,55]. Moreover, it was shown that insulin therapy facilitated re-differentiation of Ngn3-expressing endocrine progenitor cells into insulin-producing mature β-cells [55]. These findings clearly show that de-differentiation of insulin-producing mature β-cells into other cell types is involved in pancreatic β-cell failure in type 2 diabetes mellitus. Additionally, such findings show that insulin therapy protects β-cells not only through the suppression of apoptotic β-cell death, but also through the facilitation of re-differentiation of progenitor cells into insulin-producing mature β-cells.

Under diabetic conditions, oxidative stress is provoked through several pathways and is involved in pancreatic β-cell failure [76]. Since the expression levels of antioxidant enzymes in β-cells are very low compared to other tissues, it is thought that β-cells are more easily damaged by oxidative stress compared to other kinds of cells or tissues. Provoked oxidative stress reduces the expression levels of insulin and its transcription factors PDX-1 and MafA. Consequently, it is likely that chronic exposure of β-cells to a high glucose concentration finally leads to β-cell failure by inducing oxidative stress (Figure 1). Additionally, it has been shown that such a reduction in insulin biosynthesis and secretion together with a reduction in PDX-1 and MafA is preserved by mitigating pancreatic β-cell failure with insulin preparation or SGLT2 inhibitors, especially in the early stage of diabetes mellitus [86,87,88,89,90].

It has been thought that the activated JNK pathway is, at least partially, associated with β-cell failure. It was reported that inhibition of this pathway protected β-cells from oxidative stress [91]. Additionally, it was shown that inhibition of the JNK pathway suppressed nucleo-cytoplasmic translocation of PDX-1 induced by oxidative stress [92]. On the other hand, it was reported that MafA expression was not clearly observed in almost all β-cells expressing c-Jun, and that c-Jun overexpression with c-Jun expressing adenovirus in β-cells significantly reduced MafA expression level [65]. Taken together, it is likely that the activated JNK pathway and induced c-Jun expression are closely associated with β-cell failure found in type 2 diabetes mellitus (Figure 1).

Moreover, it was clearly demonstrated that MafA overexpression in β-cells preserved β-cell mass and function and finally alleviated β-cell failure, which is often observed in type 2 diabetes mellitus [67]. In β-cell-specific MafA overexpressing transgenic mice, plasma insulin levels were increased, and plasma glucose levels were decreased. Additionally, β-cell mass was preserved, and insulin biosynthesis and secretion were preserved in the β-cell-specific MafA transgenic mice [67]. In conclusion, it is likely that down-regulation of MafA expression is closely associated with β-cell failure found in type 2 diabetes mellitus.

In addition, it is known that the transcription factor Nrf2 plays a crucial role in protecting β-cells from oxidative stress. The preservation of β-cell mass and function largely depends on the presence of Nrf2. Indeed, it was reported that activated Nrf2 alleviates inflammation and maintains β-cell mass by suppressing apoptotic β-cell death and promoting β-cell proliferation. [84]. Various kinds of Nrf2 activators have been examined in clinical trials for the treatment for the preservation of β-cell function and mass in addition to the prevention of diabetic complications. We think that modulating Nrf2 activity in β-cells would be a promising and useful therapeutic approach for the treatment of type 2 diabetes mellitus.

## 3. Alteration of Exosome microRNAs in Pancreatic β-Cells Is, at Least in Part, Involved in Pancreatic β-Cell Failure Found in Type 2 Diabetes Mellitus

It has been thought that exosomes are a useful tool for the diagnosis and treatment of various diseases in the early stage of the disease. It has been shown that various kinds of exosome-microRNAs such as miR-375 and miR-29 are associated with abnormality of glucose and lipid metabolism [93,94,95]. Among them, miR-375 is closely associated with pancreatic β-cell failure. First, a combination of inflammatory cytokines induces a significant change in miR-375 expression level [96]. Second, since miR-375 expression level is higher in subjects with diabetes mellitus compared to those without it, it is likely that miR-375 is an early marker of β-cell failure. Third, miR-375 is important for glucose-regulated insulin secretion. Indeed, when human embryonic stem cells differentiate into endodermal lineages, miR-375 expression level is substantially increased. Taken together, miR-375 plays an important role in the process of pancreas development, β-cell growth and proliferation, and insulin secretion, which could be regulated by the above-mentioned pancreatic transcription factors. Furthermore, since miR-375 plays a crucial role in β-cells, it may be a potential target to treat diabetes mellitus. In addition, microRNAs secreted from β-cells can be transferred to other tissues, which in turn regulates β-cell activity. For instance, when miR-26a is transferred to the liver, it enhances insulin sensitivity and alleviates the abnormal glucose metabolism [97]. In conclusion, the alteration of exosome microRNAs in β-cells is, at least in part, involved in pancreatic β-cell failure found in type 2 diabetes mellitus.

## 4. Impairment of Incretin Signaling in Pancreatic β-Cells Is, at Least in Part, Involved in Pancreatic β-cell Failure Found in Type 2 Diabetes Mellitus

Two kinds of incretins, GLP-1 and GIP bind to each receptor in β-cells and facilitate insulin secretion. Such insulin secretion is regulated through various pathways in β-cells. First, cyclic adenosine monophosphate (cAMP) facilitates insulin secretion through phosphorylation of protein kinase A (PKA). Second, cAMP has another target Epac in β-cells [98,99,100,101]. Third, a physiologically low concentration of GLP-1 activates protein kinase C (PKC) and enhances insulin secretion [102]. Taken together, it is likely that GLP-1 enhances glucose-stimulated insulin secretion through various pathways, depending on its concentration.

It has been reported, however, that expression levels of incretin receptors in β-cells are reduced under diabetic conditions, leading to the impairment of incretin effects [103,104] (Figure 1). It has also been shown that a reduction in transcription factor 7-like 2 (TCF7L2) expression level is involved in the reduced incretin receptor expression [105,106,107]. Taken together, down-regulation of incretin receptor expression after chronic exposure to a high glucose concentration is likely associated with the impairment of incretin effects and is involved in β-cell failure found in type 2 diabetes mellitus.

It has also been reported that TCF7L2 is closely associated with the maintenance of β-cell mass and function though activation of the AKT and mTOR pathway [108,109,110,111]. Indeed, inactivation of TCF7L2 impairs insulin secretion and abnormality of glucose metabolism. Additionally, it is known that common genetic variations of TCF7L2 are associated with type 2 diabetes mellitus and that subjects with its high-risk allele of TCF7L2 show impaired insulin secretion [112,113,114,115,116].

## 5. Incretin-Based Medicine Shows Protective Effects against Pancreatic β-Cell Failure Found in Type 2 Diabetes Mellitus

Incretin-based medicines such as the GLP-1 receptor activator and DPP-IV inhibitor ameliorate glycemic control and mitigate the deterioration in β-cell function in human subjects as well as animal models. It has been reported that the GLP-1 receptor activator preserves pancreatic β-cell function and mass in several types of type 2 diabetes animals [117,118,119,120,121,122,123]. For instance, it was shown that when type 2 diabetes db/db mice were treated with the GLP-1 receptor activator, liraglutide for 2 weeks, insulin biosynthesis and glucose-stimulated insulin secretion were increased [117]. Liraglutide enhanced the gene expression involved in cellular differentiation (Hb9, NeuroD and PDX-1) and proliferation (cyclin D and Erk-1) in pancreatic islets even in normoglycemic m/m mice, strongly suggesting the direct effect of GLP-1 on β-cell kinetics [117]. There have been several similar reports so far, indicating that the GLP-1 receptor activator exerts protective effects on β-cell mass and function in other kinds of diabetic model animals [120,121,122,123]. Indeed, in alloxan-induced diabetic mice, both β-cell mass and function were substantially preserved by liraglutide treatment, which led to amelioration of glycemic control [120]. It was also shown that β-cell mass was preserved by liraglutide treatment due to an increase in β-cell proliferation and a decrease in β-cell apoptosis. Moreover, the beneficial effects of liraglutide in these mice were preserved even 2 weeks after drug withdrawal [120]. In conclusion, incretin-based medicine shows protective effects against pancreatic β-cell failure in type 2 diabetes mellitus.

It has been shown that the GLP-1 receptor activator shows more beneficial effects in the early stages of diabetes mellitus [118,119]. Obese type 2 diabetic db/db mice were treated with GLP-1 receptor activator liraglutide and/or insulin sensitizer pioglitazone for 2 weeks at 7 weeks old as the early stage and at 16 weeks old as the advanced stage. Insulin biosynthesis and glucose-stimulated insulin secretion were markedly enhanced by the treatment only in the early stage. [119]. We assume that reduced GLP-1 receptor expression after chronic exposure to a high glucose concentration explains why the GLP-1 receptor activator did not show beneficial effects in the advanced stage [119]. Taken together, we should use incretin-based medicine in the early stages without hesitation or clinical inertia in order to maintain β-cell mass and function and ameliorate glycemic control.

In addition, it was shown that DPP-IV inhibitor together with SGLT2 inhibitor exerted more favorable effects on β-cell function and mass, especially in the early stage of diabetes mellitus compared to the advanced stage in type 2 diabetic db/db mice [90]. In the study, 7-week-old and 16-week-old db/db mice were used as an early and advanced stage of diabetes mellitus, respectively, and all mice were treated for 2 weeks with DPP-IV inhibitor, linagliptin and/or SGLT2 inhibitor, empagliflozin. In the combination group, β-cell mass and function were significantly preserved compared to those without treatment only at the early stage, together with enhanced β-cell proliferation [90]. Taken together, such combination therapy shows beneficial effects on β-cells, particularly in the early stages.

## 6. GLP-1 Receptor Activator Shows Protective Effects against Pancreatic β-Cell Failure for a Long Period of Time without Down-Regulating GLP-1 Receptor Expression Level in β-Cells

In general, chronic exposure to a large amount of ligand leads to down-regulation of its receptor. Additionally, it is known that the serum GLP-1 concentration becomes extremely and non-physiologically high after usage of GLP-1 receptor activator. It remained unknown, however, whether the long-time usage of GLP-1 receptor activator down-regulates its receptor. However, it was reported that GLP-1 receptor expression was not reduced, even after treatment with the GLP-1 receptor activator, dulaglutide for as long as 17 weeks in type 2 diabetic db/db mice [124]. Treatment with dulaglutide ameliorated glycemic control for 17 weeks in the mice compared to those without treatment. In addition, treatment with dulaglutide enhanced insulin biosynthesis and glucose-stimulated insulin secretion [124]. Taken together, the GLP-1 receptor activator protects β-cells against glucose toxicity for a long time due to preservation of GLP-1 receptor expression level in β-cells.

GLP-1 binds to its receptor in various kinds of cells, and the complex of GLP-1 ligand and its receptor is internalized in the cells. It is thought that receptors that are internalized in cells preserve their expression level compared to those without internalization. Consequently, although speculative, we think that such characteristics of the GLP-1 receptor could explain, at least in part, why GLP-1 receptor expression in β-cells was not down-regulated even after long-term exposure to GLP-1 ligand. Moreover, a strategy for the use of some drugs has been developed based on such phenomena [125,126,127,128].

## 7. SARS-CoV-2 Directly Infects Pancreatic β-Cells through Neuropilin-1, Leading to Pancreatic β-Cell Failure such as Apoptotic β-Cell Death and Reduction in Insulin Secretion

Many subjects have recently suffered from life-threatening coronavirus infection, especially coronavirus-mediated pneumonia, all over the world, and coronavirus infection has brought about a new and persistent pandemic. The mortality in subjects with coronavirus infection is extremely high, and the main reason for this is coronavirus-mediated pneumonia [129]. It seems that in subjects with a coronavirus infection, various kinds of inflammatory cytokines are produced and are likely associated with the aggravation of infection. The defense mechanism against the inflammation is substantially weakened, especially in elderly subjects with comorbidities such as diabetes mellitus. Indeed, it was shown that the mortality rate due to coronavirus infection was quite high in subjects with diabetes mellitus [130,131]. Additionally, the spread of coronavirus infection has led to various limitations on the activities of daily life and the obstruction of economic development all over the world. However, we are hopeful that coronavirus infection will decline and normal daily life will soon resume all over the world

It is thought that there is some association between diabetes mellitus and coronavirus infection. Although it remains unclear, subjects with diabetes mellitus are more easily infected with coronavirus compared to healthy subjects, and it is likely that coronavirus infection becomes severe more easily in subjects with poor glycemic control compared to healthy subjects. In addition, although it has been thought that the deterioration of β-cell function is a key factor in the pathogenesis of diabetes mellitus due to coronavirus infection, it remains controversial as to whether β-cells are directly damaged by coronavirus or not. Indeed, it is thought that the coronavirus does not directly infect β-cells because of the low expression level of angiotensin-converting enzyme 2 (ACE2), which allows the coronavirus to go into cells [132,133]. Very recently, however, it has been reported that SARS-CoV-2 directly infects β-cells through neuropilin-1, leading to apoptotic β-cell death and a reduction in insulin secretion [134,135,136,137,138,139] (Figure 2). It was reported that there was SARS-CoV-2-containing nucleocapsid protein in β-cells after infection with SARS-CoV-2. It was also shown that such phenomena were suppressed by a neuropilin-1 antagonist. These data clearly indicate that neuropilin-1 is important for SARS-CoV-2 to go into β-cells. Furthermore, it was shown that neuropilin-1 expression was high in β-cells with SARS-CoV-2 infection compared to those without the infection. Infection with SARS-CoV-2 increases TUNEL-positive apoptotic β-cell death. Indeed, it was reported that infection with SARS-CoV-2 stimulated p21-activated kinase (PAK) and c-Jun N-terminal kinase (JNK) pathways in β-cells (Figure 2). Activation of the JNK pathway finally increases apoptotic β-cell death and reduces insulin secretion. In addition, as described above, it has been shown that activation of the JNK pathway reduces the expression level and activity of insulin gene transcription factor PDX-1, which we assume also leads to a reduction in insulin biosynthesis and secretion. It was also shown that the expression of α-cell markers and acinar cell markers was increased in β-cells after SARS-CoV-2 infection. Therefore, it is possible that β-cells undergo trans-differentiation to α-cells or acinar cells after the infection. It was also reported that eIF2-mediated response was closely associated with the pathology of β-cell failure induced by SARS-CoV-2 infection. (Figure 2). In conclusion, it is likely that SARS-CoV-2 directly infects pancreatic β-cells through neuropilin-1, and various pathways are activated in β-cells, and finally, apoptotic β-cell death, a reduction in insulin secretion, and trans-differentiation of β-cells into other cell types are brought about by the activation of various pathways in β-cells (Figure 2).

## 8. Conclusions

Various transcription factors play crucial roles in the differentiation of endocrine progenitor cells into mature insulin-producing β-cells and preservation of adult β-cell function. However, after the exposure of β-cells to a high glucose concentration for a long period of time under diabetic conditions, the expression levels and activities of PDX-1 and MafA are reduced, which leads to β-cell failure. Additionally, the expression levels of incretin receptors in β-cells are reduced after the onset of diabetes mellitus. It is likely that the reduced expression level of insulin gene transcription factors and incretin receptors explains, at least in part, the molecular mechanism for β-cell failure found in type 2 diabetes mellitus. Additionally, since incretin receptor expression is reduced in the advanced stage of diabetes mellitus, incretin-based medicine shows more favorable effects against β-cell glucose toxicity, especially in the early stage of diabetes mellitus compared to the advanced stage. On the other hand, many subjects have recently suffered from life-threatening coronavirus infection and coronavirus infection has brought about a new and persistent pandemic. Additionally, coronavirus infection has led to various limitations on daily activities and restricted economic development worldwide. It has been reported recently that SARS-CoV-2 directly infects β-cells through neuropilin-1, leading to apoptotic β-cell death and reduction in insulin secretion. Additionally, it was shown that there was SARS-CoV-2-containing nucleocapsid protein in β-cells after coronavirus infection. In this review article, we featured a possible molecular mechanism for pancreatic β-cell failure, which is often observed in type 2 diabetes mellitus. Finally, we are hopeful that coronavirus infection will be cleared up and normal daily life will soon resume all over the world.

## Figures and Tables

**Figure 1 biomedicines-10-00818-f001:**
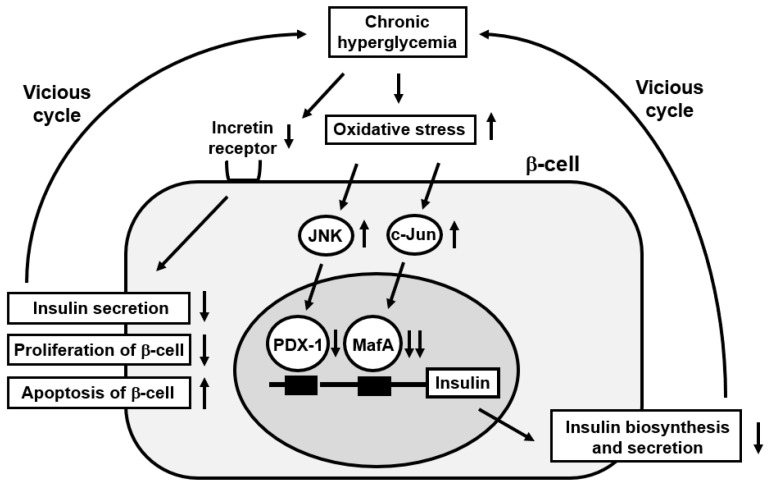
Possible underlying pathway of β-cell failure. Chronic hyperglycemia provokes oxidative stress and thus substantially reduces PDX-1 and MafA expression in nuclei, which finally reduces insulin biosynthesis and secretion. After chronic exposure to a high glucose concentration, incretin receptor expression level is also reduced, which leads to pancreatic β-cell failure.

**Figure 2 biomedicines-10-00818-f002:**
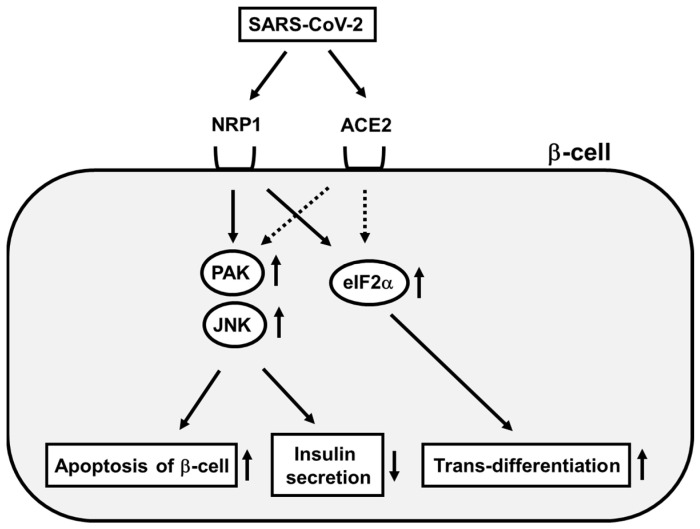
Pancreatic β-cell failure induced by coronavirus infection. SARS-CoV-2 directly binds to NRF1 and ACE2 in pancreatic β-cell membrane. Then, PAK, JNK and eIF2α are activated within pancreatic β-cells, which finally leads to increase in apoptotic β-cell death, reduction in insulin biosynthesis, and trans-differentiation of β-cells to other cell types.

**Table 1 biomedicines-10-00818-t001:** Expression pattern in mature islets and phenotype in the pancreas in knockout mice of each pancreatic transcription factor.

Transcription Factor	Expression Site in Mature Islets	Pancreas-Related Phenotype in Each Knockout Mouse
PDX-1	β- and δ-cells	absence of the pancreas
Hb9	β-cells	absence of the dorsal pancreas
Isl-1	all islet cells	absence of islet cells and dorsal pancreatic mesoderm
Pax4	not detected	absence of β- and δ-cells increase in α- and ε-cells
Pax6	all islet cells	absence of α-cells decrease in β-, δ- and PP-cells increase in ε-cells
Nkx2.2	α-, β- and PP-cells	absence of β-cells decrease in α- and PP-cells
Nkx6.1	β-cells	decrease in β-cells
Ngn3	not detected	absence of endocrine cells
NeuroD	all islet cells	decrease in endocrine cells
MafA	β-cells	decrease in insulin biosynthesis and secretion

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
