# Peer review of "Molecular Mechanism of Pancreatic β-Cell Failure in Type 2 Diabetes Mellitus"

_biomedicines, 2022, doi:10.3390/biomedicines10040818_

Round 1

Reviewer 1 Report

In the review submitted by Hideaki Kaneto et al, the authors summarized the possible molecular mechanism of pancreatic β-cell failure in type-2 Diabetes Mellitus (T2DM). Here the detailed roles of transcription factors PDX-1, MafA, and incretin receptors are majorly discussed in the decline of β-cell which has been recognized as the key contributing factor to the progression of T2DM.

Here is a major concern:

T2DM is more likely caused by a set of unique metabolic stressors to the β cell including elevated glucose or increased free fatty acids. There are tight connections between metabolic syndrome and T2DM progression. Since T2DM is mediated by metabolic mechanisms, it would be better to add a few paragraphs to discuss the underlying molecular mechanisms at the initiation stage of T2DM such as the responses of β-cell to metabolism stressors.

Minor Concerns:

There is a missing “.” in line 191.

There is a missing space in line 214 between GLP-1 and the receptor.

There is a missing “.” in line 231.

Author Response

In the review submitted by Hideaki Kaneto et al, the authors summarized the possible molecular mechanism of pancreatic β-cell failure in type-2 Diabetes Mellitus (T2DM). Here the detailed roles of transcription factors PDX-1, MafA, and incretin receptors are majorly discussed in the decline of β-cell which has been recognized as the key contributing factor to the progression of T2DM.

Here is a major concern:

T2DM is more likely caused by a set of unique metabolic stressors to the β cell including elevated glucose or increased free fatty acids. There are tight connections between metabolic syndrome and T2DM progression. Since T2DM is mediated by metabolic mechanisms, it would be better to add a few paragraphs to discuss the underlying molecular mechanisms at the initiation stage of T2DM such as the responses of β-cell to metabolism stressors.

Thank you very much for your valuable suggestion. According to your kind suggestion, we added the description about this point in the revised version of the manuscript (page 3, lines 84-97)

Minor Concerns:

There is a missing “.” in line 191.

There is a missing space in line 214 between GLP-1 and the receptor.

There is a missing “.” in line 231.

Thank you very much. We amended these points in the revised version

Thank you very much again for your thoughtful comments that have led to strengthening our manuscript.

Reviewer 2 Report

The review work presented by Hideaki Kaneto and colleagues entitled “Molecular Mechanism of Pancreatic β-cell Failure in Type 2 Diabetes Mellitus” is well written, clear, and easy to read. The topic is interesting and therefore, it adds clustered information to the area o f beta-cell dysfunction. 

Please add a section on microRNAs see this recent work for this concern.

Pharmaceuticals (Basel). 2021 Dec 2;14(12):1257. doi: 10.3390/ph14121257.

Author Response

The review work presented by Hideaki Kaneto and colleagues entitled “Molecular Mechanism of Pancreatic β-cell Failure in Type 2 Diabetes Mellitus” is well written, clear, and easy to read. The topic is interesting and therefore, it adds clustered information to the area of beta-cell dysfunction. 

Please add a section on microRNAs see this recent work for this concern.

Pharmaceuticals (Basel). 2021 Dec 2;14(12):1257. doi: 10.3390/ph14121257.

Thank you very much for your valuable suggestion. According to your kind suggestion, we added thesection about microRNAs in the revised version of the manuscript (page 5, lines 172-192).

Thank you very much again for your thoughtful comments that have led to strengthening our manuscript.
